# A Genetically Engineered Bivalent Vaccine Coexpressing a Molecular Adjuvant against Classical Swine Fever and Porcine Epidemic Diarrhea

**DOI:** 10.3390/ijms241511954

**Published:** 2023-07-26

**Authors:** Hao Wang, Weicheng Yi, Huan Qin, Qin Wang, Rui Guo, Zishu Pan

**Affiliations:** 1State Key Laboratory of Virology, College of Life Sciences, Wuhan University, Wuhan 430072, China; wanghao1993@whu.edu.cn (H.W.); yiweicheng@whu.edu.cn (W.Y.); qinhuan2100@163.com (H.Q.); 2World Organization for Animal Health Reference Laboratory for Classical Swine Fever, China Institute of Veterinary Drug Control, Beijing 100081, China; wq551@vip.sina.com; 3Key Laboratory of Prevention and Control Agents for Animal Bacteriosis (Ministry of Agriculture and Rural Affairs), Hubei Provincial Key Laboratory of Animal Pathogenic Microbiology, Institute of Animal Husbandry and Veterinary, Hubei Academy of Agricultural Sciences, Wuhan 430064, China; hlguorui@163.com

**Keywords:** classical swine fever virus, porcine epidemic diarrhea virus, bivalent vaccine, VISA, molecular adjuvant

## Abstract

Classical swine fever (CSF) and porcine epidemic diarrhea (PED) are highly contagious viral diseases that pose a significant threat to piglets and cause substantial economic losses in the global swine industry. Therefore, the development of a bivalent vaccine capable of targeting both CSF and PED simultaneously is crucial. In this study, we genetically engineered a recombinant classical swine fever virus (rCSFV) expressing the antigenic domains of the porcine epidemic diarrhea virus (PEDV) based on the modified infectious cDNA clone of the vaccine strain C-strain. The S1N and COE domains of PEDV were inserted into C-strain cDNA clone harboring the mutated 136th residue of N^pro^ and substituted 3′UTR to generate the recombinant chimeric virus vC/SM3′UTR_N_-S1NCOE. To improve the efficacy of the vaccine, we introduced the tissue plasminogen activator signal (tPAs) and CARD domain of the signaling molecule VISA into vC/SM3′UTR_N_-S1NCOE to obtain vC/SM3′UTR_N_-tPAsS1NCOE and vC/SM3′UTR_N_-CARD/tPAsS1NCOE, respectively. We characterized three vaccine candidates in vitro and investigated their immune responses in rabbits and pigs. The N^pro^_D136N_ mutant exhibited normal autoprotease activity and mitigated the inhibition of IFN-β induction. The introduction of tPAs and the CARD domain led to the secretory expression of the S1NCOE protein and upregulated IFN-β induction in infected cells. Immunization with recombinant CSFVs expressing secretory S1NCOE resulted in a significantly increased in PEDV-specific antibody production, and coexpression of the CARD domain of VISA upregulated the PEDV-specific IFN-γ level in the serum of vaccinated animals. Notably, vaccination with vC/SM3′UTR_N_-CARD/tPAsS1NCOE conferred protection against virulent CSFV and PEDV challenge in pigs. Collectively, these findings demonstrate that the engineered vC/SM3′UTR_N_-CARD/tPAsS1NCOE is a promising bivalent vaccine candidate against both CSFV and PEDV infections.

## 1. Introduction

Classical swine fever (CSF) and porcine epidemic diarrhea (PED) are responsible for heavy economic losses to the swine industry worldwide due to their high morbidities and mortalities. CSF is characterized by fever, hemorrhage, and leukopenia caused by the CSF virus (CSFV) [1,2]. PED is characterized by vomiting, diarrhea, and dehydration and is caused by a porcine coronavirus, the PED virus (PEDV) [3,4].

CSFV, belonging to the *Pestivirus* genus within the family *Flaviviridae* [5], possesses an approximately 12.3 kb single-stranded, positive-sense RNA genome. The CSFV genome harbors a large open-reading frame (ORF) wedged between the 5′UTR and 3′UTR [6] and encodes a 3898-amino acid polyprotein, which can be co- and post-translationally split by viral and host proteases into four structural proteins and eight nonstructural proteins [7,8,9]. N^pro^ is an autoprotease located at the N-terminus of the polyprotein and disrupts type I IFN (IFN-α/β) synthesis by inducing IRF3 degradation, which endows CSFVs with high antivirus immune escape abilities [10,11,12,13,14]. To control this Class-A infectious disease classified by the World Organization for Animal Health (OIE), the CSFV C-strain vaccine has been regarded as one of the safest and most effective attenuated vaccines for CSF prevention [15,16,17]. Furthermore, the C-strain has been used to successfully express foreign proteins [18,19], indicating that it is a potential vaccine vector.

PEDV is another important pathogen that harms pigs, especially suckling piglets, attracting widespread attention. PEDV is a member of the genus *Alphacoronavirus* within the family *Coronaviridae* [20]. With approximately 28 kb, the positive-stranded RNA genome contains seven ORFs that encode 4 structural proteins and 16 nonstructural proteins [21]. The spike (S) protein, consisting of S1 and S2 domains [22], mediates viral cell entry and serves as a major target for neutralizing antibodies [23,24]. Meanwhile, most neutralizing antibodies against coronaviruses inhibit viral entry by interfering with the receptor-S1 domain [25,26]. Several neutralizing domains, such as the ‘‘N-terminal sialic acid binding domain” (S1^0^ or S1N) and ‘‘receptor binding domain” (S1^B^ or COE) regions, were identified in S1 [24,27,28]. Remarkably, since 2010, the highly pathogenic and rapidly spreading PEDV variant (GII) emerging in China has kept challenging traditional PEDV vaccines such as CV777 [29,30]. However, the S1N domain is relatively conserved in the PEDV GII strains, which could be a promising protective epitope to control PEDV GII strain infection [27,31]. Therefore, the fused S1NCOE represents a candidate subunit vaccine for PED prevention.

Traditional CSFV and PEDV attenuated vaccines are separately inoculated more than eight times a year to induce high production of protective maternal antibodies in sows on many farms, which is time-consuming and expensive [32,33]. Bivalent vaccines against CSF and PED have the potential to solve these problems. However, there are no licensed bivalent vaccines yet. Moreover, the emergence of the highly pathogenic and rapidly spreading PEDV variant (GII) makes traditional PEDV vaccines inefficient. Therefore, it is urgent to develop an effective bivalent vaccine against both CSFV and PEDV. Adjuvants help to improve vaccine efficacy and molecular adjuvants have drawn growing attention due to the increasingly detailed description of antivirus immune pathways [34]. Previous reports indicated that the caspase recruitment domain (CARD) of VISA coexpression can trigger innate intracellular immune signaling to promote vaccine immunogenicity [35], which could be a promising molecular adjuvant.

Here, we constructed a chimeric CSFV C-strain expressing the fused S1NCOE protein of the virulent PEDV GII strain GX750A. Meanwhile, coexpression of the CARD domain of VISA was employed to fortify bivalent vaccine efficacy. Furthermore, we characterized these constructs in vitro and investigated the immune protection against virulent CSFV and PEDV in rabbits and pigs.

## 2. Results

### 2.1. Amino Acid Substitution D_136_N in N^pro^ of CSFV C-Strain Inhibited IRF3 Degradation and Upregulated VISA-Mediated IFN-β Induction

N^pro^ disrupts type I IFN (IFN-α/β) synthesis by inducing IRF3 degradation and suppressing IRF1 expression and nuclear translocation to reduce type III IFN (IFN-λ) production [13,36]. N^pro^ participates in antiviral immune escape, which may limit the immune benefit after vaccination. Previous studies demonstrated that residue Asp136 of N^pro^ plays a crucial role in inducing the degradation of IRF3 and suppressing IFN-β production [12].

To verify the effect, we substituted the 136th Asp into Asn in N^pro^ of the CSFV C-strain and checked the degradation of IRF3 and suppression of IFN-β induction. For a better understanding, we used N^pro^ of the highly virulent CSFV Shimen strain as a positive control (termed N^pro^-SM). As indicated, N^pro^-SM and N^pro^ of C-strain (N^pro^-C) overexpression reduced IRF3 expression to undetectable levels while IRF3 remained under 136th Asp-substituted N^pro^-C (N^pro^_D136N_-C) overexpression (Figure 1A). In the VISA-overexpressing swine kidney SK-6 cell line, N^pro^-SM and N^pro^-C significantly decreased IFN-β promoter activation and 136th amino acid substitution and largely restored the suppression accordingly (Figure 1B). It seems that C-strain wild-type N^pro^ plays a critical role in antivirus immune responses, comparable to the Shimen strain.

As previously reported, we generated C-strain chimera with the 3′UTR of the Shimen strain for better virus amplification productivity (termed as vC/SM3′UTR) [37]. To confirm these observations, SK-6 cells were infected with different recombinant CSFV strains, and the highly virulent CSFV Shimen strain displayed dramatic suppression of IRF3 expression (Figure 1C). Compared to the wild-type C-strain virus-infected group, IRF3 degradation was more moderate once Asp136 of N^pro^ was substituted (vC/SM3′UTR_N_) (Figure 1C). In the VISA-overexpressing swine kidney SK-6 cell line, the CSFV Shimen strain inhibited IFN-β transcription the most, and vC/SM3′UTR_N_ exhibited the weakest effects compared with the other CFSV strains (Figure 1D).

Similar to the Shimen strain, we demonstrated that the 136th Asp of N^pro^ from the well-accepted vaccine C-strain is also necessary for antiviral innate immune inhibition effects. D_136_N substitution significantly reversed the immune suppression by N^pro^, which may promote the vaccine response. Nevertheless, whether the D_136_N substitution affects N^pro^ autoprotease function remains unknown.

### 2.2. Expression of N^pro^ or Its Variant Fused with CARD Domain of VISA and Activation of the IFN-β Promoter

To improve the efficacy of vectored CSFV vaccines, we investigated whether CARD fused with N^pro^ or with its variant affects IFN type I induction. Plasmids expressing N^pro^, N^pro^_D136N_, N^pro^CARD, and N^pro^_D136N_CARD with a Flag tag were constructed (Figure 2A). SK-6 cells transfected with the indicated plasmids were used to detect IFN-β promoter activity. The results showed that the fusion protein N^pro^CARD activated the IFN-β promoter, and that N^pro^_D136N_CARD exhibited significantly increased IFN-β induction compared to N^pro^CARD (Figure 2B). The IFA results indicated that N^pro^_D136N_CARD was located in the cytoplasm and nucleus of transfected PK-15 cells (Figure 2C). To investigate the effect of amino acid substitution D_136_N and insertion of CARD on the proteolytic activity of N^pro^, we constructed plasmids pVAX1-EGFP, pVAX1-N^pro^-CoreEGFP, pVAX1-N^pro^_D136N_-CoreEGFP, and pVAX1-N^pro^_D136N_CARD-CoreEGFP (Appendix A). Western blotting identified that the amino acid substitution D_136_N, and the insertion of CARD did not affect the autoprotease activity of N^pro^ (Figure 2D). The findings suggest that the N^pro^_D136N_CARD protein could be correctly processed and function in VISA-mediated IFN-β induction and N^pro^ autoprotease activity. Taken together, CARD coexpression and D136N substitution have a synergistic effect in inducing IFN-β expression without disrupting the autoprotease function of N^pro^.

### 2.3. Generation and Identification of Recombinant CSFVs Expressing CARD and S1NCOE Peptides

To generate recombinant vaccine candidates against CSFV and PEDV, we constructed three chimeric infectious CSFV cDNA clones harboring S1NCOE-, tPAsS1NCOE-, or CARD/tPAsS1NCOE-encoded fragments. An infectious CSFV cDNA clone with a substitution of the Shimen 3′UTR was used as the parental control (Figure 3A). Transfected PK-15 cells were serially passaged three times, and the rescue of recombinant CSFVs was identified by IFA and Western blotting. The expression of foreign proteins was confirmed by Western blotting. The data showed that the PK-15 cells transfected with all four infectious cDNA clones were stained with anti-NS3 antibody and that the PK-15 cells transfected with cDNA clones harboring S1NCOE could be stained with anti-PEDV S1 (Figure 3B). Western blotting confirmed the expression of both the S1NCOE and CARD peptides and that fusion with the tPA signal resulted in the secretory expression of the S1NCOE peptide (Figure 3C). These results demonstrated that three recombinant CSFVs expressing different forms of foreign peptides, vC/SM3′UTR_N_-S1NCOE, vC/SM3′UTR_N_-tPAsS1NCOE, and vC/SM3′UTR_N_-CARD/tPAsS1NCOE, were successfully rescued.

### 2.4. Characterization of Recombinant CSFVs Expressing CARD and S1NCOE Peptides

Recombinant CSFVs expressing foreign protein were further characterized. Growth kinetic analysis showed that a similar growth kinetic curve was observed for the three recombinant CSFVs and that recombinant CSFVs expressing foreign peptides resulted in slightly reduced peak titers compared to the parental vC/SM3′UTR (Figure 4A). As expected, vC/SM3′UTR_N_-CARD/tPAsS1NCOE significantly increased IFN-β promoter induction (Figure 4B). Meanwhile, all recombinant CSFVs formed similar plaque sizes in SK-6 cells (Figure 4C).

To evaluate the stability of chimeric viruses, recombinant CSFVs were continuously passaged on PK-15 cells for 30 generations. The expressed foreign proteins of the 30th (P30) passaged recombinant CSFVs were detected by IFA (Figure 4D) and Western blotting (Figure 4E). The data showed that the S1NCOE and CARD peptides were effectively expressed in PK-15 cells that had been infected with the 30th passaged recombinant CSFVs (Figure 4D,E). Sequencing of amplified fragments from the viral genome confirmed the stability of the inserted gene. Viral titrations showed that as a bivalent vaccine candidate, the representative vC/SM3′UTR_N_-CARD/tPAsS1NCOE from the 1st, 10th, 20th, and 30th generations exhibited similar titers (Figure 4F). Our results demonstrated that the growth characteristics of the passaged recombinant CSFVs were unchanged and that the heterogeneous peptides S1NCOE and CARD could be stably expressed.

### 2.5. Humoral Responses and PEDV-Specific IFN-γ Production Induced by the Recombinant CSFVs in Experimental Animals

To evaluate humoral and cellular responses induced by recombinant CSFVs in experimental animals, we first measured CSFV- and PEDV-specific antibody titers in the serum of rabbits inoculated with four recombinant CSFVs at 14 days postbooster. The data showed that similar titers of CSFV-specific antibodies (Figure 5A) and CSFV-neutralizing antibodies (Figure 5B) were elicited by the four recombinant CSFVs. As expected, PEDV-specific antibodies (Figure 5C) and PEDV-neutralizing antibodies (Figure 5D) were observed only in the serum of rabbits inoculated with recombinant CSFVs expressing the S1NCOE peptide (Figure 5C,D). Compared to that with vC/SM3′UTR_N_-S1NCOE, inoculation with vC/SM3′UTR_N_-tPAsS1NCOE or vC/SM3′UTR_N_-CARD/tPAsS1NCOE significantly increased PEDV antibody responses (*p* < 0.001) (Figure 5C,D). However, coexpression of the CARD domain of VISA did not influence the production of PEDV-specific antibodies and virus-neutralizing antibodies (*p* > 0.05) (Figure 5C,D). Cytokine analysis showed that inoculation with vC/SM3′UTR_N_-CARD/tPAsS1NCOE significantly increased PEDV-specific IFN-γ secretion compared to vC/SM3′UTR_N_-S1NCOE or vC/SM3′UTR_N_-tPAsS1NCOE. Coexpression of the CARD domain of VISA significantly upregulated IFN-γ secretion (*p* < 0.001) (Figure 5E).

We further investigated immune responses in pigs induced by the representative recombinant virus vC/SM3′UTR_N_-CARD/tPAsS1NCOE. Similar CSFV-specific antibodies (Figure 6A) and CSFV-neutralizing antibodies (Figure 6B) were detected in the serum of piglets vaccinated with vC/SM3′UTR_N_-CARD/tPAsS1NCOE and compared with those vaccinated with the CSFV C-strain vaccine. From 14 to 28 days post-vaccination (dpv), decreased PEDV-specific IgG (Figure 6C) and PEDV-neutralizing antibody (Figure 6D) titers were observed in the serum of piglets vaccinated with vC/SM3′UTR_N_-CARD/tPAsS1NCOE compared with inactivated PEDV. However, higher PEDV-specific IgA levels were elicited in the vC/SM3′UTR_N_-CARD/tPAsS1NCOE-vaccinated groups (Figure 6E). Importantly, a significantly increased specific IFN-γ concentration was detected in the supernatant of peripheral blood mononuclear cell (PBMC) cultures from piglets vaccinated with vC/SM3′UTR_N_-CARD/tPAsS1NCOE (*p* < 0.01) (Figure 6F).

### 2.6. Vaccination with vC/SM3′UTR_N_-CARD/tPAsS1NCOE Protected Pigs against Virulent CSFV and PEDV Challenge Infection

The properties of the bivalent vaccine vC/SM3′UTR_N_-CARD/tPAsS1NCOE were further investigated in experimental pigs. Having been vaccinated with vC/SM3′UTR_N_-CARD/tPAsS1NCOE or C-strain vaccine, none of the piglets displayed fever or other clinical signs (Figure 7A,B). Similar peripheral White blood cell (WBC) and platelet (PLT) counts were detected in both groups (Figure 7C,D), suggesting that vC/SM3′UTR_N_-CARD/tPAsS1NCOE was completely avirulent for piglets.

After lethal CSFV challenge infection, the piglets vaccinated with vC/SM3′UTR_N_-CARD/tPAsS1NCOE were in good condition and did not exhibit fever and clinical symptoms of CSF, which was consistent with those in piglets vaccinated with the C-strain (Figure 7A,B). The peripheral WBC and PLT counts did not change (Figure 7C,D). All vaccinated piglets survived, with no viral RNA copies detected in the blood samples or oral and fecal swabs (Table 1). In contrast, DMEM inoculation induced fever and typical symptoms of CSF in piglets (Figure 7A,B) and significantly reduced WBC and PLT counts after lethal CSFV challenge (Figure 7C,D). All DMEM-inoculated piglets died within 11 days post-challenge (dpc), while CSFV RNA was detected in blood, oral, and fecal swabs of the piglets in the DMEM-inoculated group upon virulent CSFV challenge infection (Table 1). However, no viruses appeared in fecal and tissue samples of vC/SM3′UTR_N_-CARD/tPAsS1NCOE-vaccinated piglets after lethal CSFV challenge (Table 2).

The vaccinated piglets were challenged with an oral 5 × 10^5^ TCID_50_ virulent PEDV strain 14 days after the booster. The piglets vaccinated with vC/SM3′UTR_N_-CARD/tPAsS1NCOE or inactivated PEDV demonstrated normal appetites and feces with a lower density score (Table 3). However, the DMEM-inoculated piglets developed clinical PED signs, with lethargic and higher fecal scores. Compared to DMEM-inoculated piglets, transiently low PEDV RNA copies were detected in the vaccinated piglets (Table 3). Viral RNA copy numbers are widely used to determine virus concentration (typically around 100-fold higher than TCID_50_) in tissues or swabs and are considered adequate for assessing vaccine protection efficacy [38,39]. In summary, vC/SM3′UTR_N_-CARD/tPAsS1NCOE is avirulent for piglets, and its vaccination confers effective protection against virulent PEDV challenge in pigs. Our results suggest that vC/SM3′UTR_N_-CARD/tPAsS1NCOE is a promising bivalent vaccine candidate for the swine industry.

## 3. Discussion

Both CSF and PED are highly contagious and cause lethal diseases in piglets, posing a serious threat to the swine industry. It is highly advantageous to develop a bivalent vaccine that could simultaneously protect against both CSF and PED. The CSF vaccine C-strain is a pivotal measure in the control of the CSF epidemic [15,40] and is exploited as a viral vector in the development of bivalent vaccines for swine diseases [18,41].

PEDV can be classified into genotypes GI and GII based on phylogenetic analysis of the S gene [42], which can be further subdivided into five subgroups (GI-a, GI-b, GII-a, GII-b, and GII-c) [43]. In this study, the GX750A strain, which belongs to the GII-a subgroup, was utilized. The S protein of GX750A strain exhibits two insertion regions (^59^QGVN^62^ and ^140^N) and one deletion region (^161^GK^162^) within the N-terminal region, distinguishing it from the classical GI strain [44]. These insertion and deletion mutations caused variability in viral genome and may enhance pathogenicity and antigenicity in PEDV GII variants [45]. Currently, the majority of the epidemic PEDV strains in China in recent years have belonged to the GII genogroup, with the GII-a subgroup emerging as the dominant subgroup, causing epidemic outbreaks in swine population [46]. Consequently, previous commercial attenuated and inactivated PEDV vaccines based on GI strains have lost their effectiveness due to the genetic mutations, leading to continued PEDV epidemics in recent years [29,33]. Therefore, it is necessary and urgent to develop new vaccines targeting the virulent PEDV GII variants. Generating a live attenuated vaccine from a reverse genetic system has been demonstrated as a promising strategy to develop coronavirus vaccines [47,48,49]. In this study, we generated recombinant C-strain viruses that express the protective epitopes S1N and COE of the PEDV GII strain as a bivalent vaccine. Given the prevailing circumstances in the pig industry, immunization with the C-strain is crucial for effectively preventing and controlling CSFV outbreaks. In this context, our vaccine candidate holds the distinct advantage of simplifying existing immunization protocols, as a single immunization can effectively provide protection against CSFV and PEDV. Meanwhile, a subset of the neutralizing antibodies induced by the COE epitope were broadly reactive, which could target both GI and GII genotypes [27]. Therefore, future studies are needed to evaluate the ability of this vaccine candidate to provide protection against GI strain and compare it with previous PEDV GI vaccines, such as CV777 and DR13. Furthermore, the antigen expression form is closely related to inducing immune efficacy by influencing antigen localization and presentation by class II major histocompatibility [50,51]. A heterologous tPAs sequence could drive a target protein into the cellular secretion pathway [52]. We found that the chimeric C-strain virus expressing the tPAsS1NCOE fusion protein exhibited the secretory expression of S1NCOE in infected cells. Vaccination with vC/SM3′UTR_N_-tPAsS1NCOE induced a higher PEDV-specific antibody level than vaccination with vC/SM3′UTR_N_-S1NCOE in rabbits.

Type I interferons have been shown to be potent vaccine adjuvants and are capable of inducing IgG2a and IgA production and conferring protection against virus infection [53]. An adaptor molecule in the RIG-I pathway, a virus-induced signaling adaptor (VISA), activates NF-κB and IRF3 to produce type I IFN [54,55]. The CARD domain of VISA is essential for signal transmission by homotypic interactions with the CARDs of upstream RIG-I-like helicases [56]. The coexpression of VISA can regulate balanced immune responses and enhance immune protection against virus infection induced by antigens [35,57,58]. In this study, we employed an optimized vaccine design strategy to upgrade the efficacy of the bivalent vaccine. We explored the effect of the 136th residue substitution (D136N) of the N^pro^ protein on IFN-β induction and the VISA CARD as a potential genetic adjuvant of the PEDV subunit vaccine. Our results demonstrated that the N^pro^-mediated degradation of IRF3 and inhibition of IFN-β induction were abrogated by substituting the 136th residue (D136N) of the N^pro^ protein (Figure 1), and the coexpression of VISA CARD significantly upregulated IFN-β activation (Figure 4). In vaccinated piglets, immunization with vC/SM3′UTR_N_-CARD/tPAsS1NCOE not only elicited robust PEDV-specific IgG and IgA, but also induced a high IFN-γ concentration, which showed that viral shedding and clinical signs were significantly reduced after virulent PEDV challenge. Nevertheless, in future studies, vC/SM3′UTR_N_-CARD/tPAsS1NCOE should be applied to immunize pregnant sows to determine whether it is capable of eliciting lactogenic immunity that produces sIgA antibody secretion in the colostrum and milk and is transferred to suckling neonatal piglets to protect against PEDV.

In conclusion, our results demonstrated that vC/SM3′UTR_N_-tPAsS1NCOE containing tPAs exhibited secretory expression of S1NCOE in infected cells and that a higher PEDV-specific antibody level was observed in vaccinated rabbits. Among recombinant viruses, vaccination with vC/SM3′UTR_N_-CARD/tPAsS1NCOE induced a high immune response in piglets and conferred protection against virulent CSFV and PEDV infection. Taken together, vC/SM3′UTR_N_-CARD/tPAsS1NCOE expressing the genetic adjuvant CARD domain and secretory S1NCOE is a promising bivalent vaccine against CSFV and PEDV infection.

## 4. Materials and Methods

### 4.1. Cells and Viruses

Swine kidney-6 (SK-6), porcine kidney-15 (PK-15), and Vero cells were cultured in Dulbecco’s modified Eagle medium (DMEM; Invitrogen, Carlsbad, CA, USA) containing 10% fetal bovine serum (FBS; Gibco, Carlsbad, CA, USA) in a 37 °C cell culture incubator. The CSFV Shimen (SM) strain (GenBank accession number AF092248.2) and C-strain (GenBank accession number AY805221) were generated from the cDNA clones pSPT_I_/SM and pSPT_I_/C, respectively [59]. Porcine epidemic diarrhea virus (PEDV) virulent GII strain GX750A (GenBank accession number KY793536.1) was preserved in our laboratory. The chimeric infectious cDNA clone pC/SM3′UTR was constructed as described previously [37].

### 4.2. Construction of Recombinant Plasmids

The fragment encoding the C-strain N^pro^ protein was amplified by PCR using the CSFV cDNA clone pSPTI/C [59] as a template with the specific primers N^pro^(C)-F and N^pro^-Flag-R. The amplified product was cloned into pVAX1 to generate pVAX1-N^pro^-Flag using the restriction enzymes *Afl* II and *Not* I. N^pro^_D136N_ harboring the D136N mutation was introduced by PCR-based site-directed mutagenesis using pVAX1-N^pro^-Flag as a template with the specific primers D136N-F/D136N-R. The *Afl* II/*Not* I-digested N^pro^_D136N_ fragment was cloned into pVAX1 to generate pVAX1-N^pro^_D136N_-Flag. The porcine VISA gene (GenBank accession number XM_005672763) and the encoding sequence of porcine teschovirus 1 2A peptide (2A) (GenBank accession number NC_003985) were synthesized by Sangon Biotech (Shanghai, China). To obtain the fused N^pro^CARD fragment, an insertion of CARD between aa 13 and 14 of N^pro^, the 1–39 nt sequence of N^pro^ was fused with the CARD gene, using porcine VISA as a template with primers N^pro^CARD-F and CARD-R. The 40–504 nt sequence of N^pro^ was amplified using pVAX1-N^pro^-Flag as a template with primers CARD-F and N^pro^CARD-R. Then, the two fragments were fused through overlap extension PCR with primers N^pro^CARD-F and N^pro^CARD-R to obtain the N^pro^CARD fragment. The N^pro^CARD fragment was cloned into pVAX1 to construct pVAX1-N^pro^CARD-Flag. The pN^pro^_D136N_CARD was obtained by PCR-based site-directed mutagenesis using pVAX1-N^pro^CARD-Flag as a template. Similarly, the pN^pro^-CoreEGFP, pN^pro^_D136N_-CoreEGFP, and pN^pro^_D136N_CARD-CoreEGFP plasmids were constructed as described (Appendix A). All constructs were verified by sequencing.

### 4.3. Construction of Chimeric cDNA Clones and Virus Rescue

To construct the cDNA clone (pC/SM3′UTR_N_) of the C-strain harboring the D136N mutation and 3′UTR substitution, the N^pro^_D136N_ fragment (F_A_) harboring the D136N mutation was amplified by PCR using the plasmid pVAX1-N^pro^_D136N_-Flag as a template with the specific primers C292-F/C901-R. Then, the fragment (F_B_) covering the core to E2 region (nucleotides 878 to 3243) was amplified by PCR using pSPT_I_/C cDNA [59] as a template with primers C878-F/C3243-R (Appendix A). After purification of PCR products, fragment F_C_ was obtained by overlap extension PCR using a mixture of fragments A and B as templates with primers C292-F/C3243-R. After digestion with the restriction enzymes *Pml* I and *Nsi* I, the digested F_C_ fragment was cloned into *Pml* I/*Nsi* I-digested pSPT_I_/C/SM3′UTR cDNA [37] to generate the infectious cDNA clone pC/SM3′UTR_N_ (Appendix A).

To construct the infectious cDNA clone expressing the foreign epitopes pC/SM3′UTR_N_-S1NCOE, the expression cassette S1NCOE/2A (encoding sequence 7-aa (SDDGASG) in the N-terminus of the CSFV core—S1N (22–247 aa)—[G_4_S]_3_ linker—COE (499−638 aa)—porcine teschovirus 1 2A (ATNFSLLKQAGDVEENPGP)) was synthesized by Sangon Biotech (Shanghai, China). The synthesized S1NCOE/2A fragment was inserted between N^pro^ and the core genes of pC/SM3′UTR_N_ to generate pC/SM3′UTR_N_-S1NCOE (Figure 3A) by a series of steps (Appendix A). The tPAs/S1NCOE/2A fragment was obtained by introduction of the tPAs into S1NCOE/2A using overlap PCR with primers tPAs-F/tPAs-R and tPAsS1N-F/2A-R (Appendix A). Then, the tPAs/S1NCOE/2A fragment was cloned into *Pml* I/*Nsi* I-digested pC/SM3′UTR to generate pC/SM3′UTR_N_-tPAsS1NCOE. Similarly, the fragment CARD/tPAsS1NCOE/2A was cloned into *Pml* I/*Nsi* I-digested pC/SM3′UTR to generate pC/SM3′UTR_N_-CARD/tPAsS1NCOE (Figure 3A and Appendix A). All constructs of chimeric cDNA clones were sequenced to confirm the identities.

The recombinant virus was generated from complementary DNA (cDNA), following a previously described method [59]. In brief, a mixture containing 2.5 μg of cDNA and 5 μL of Lipofectamine^®^3000 (Invitrogen, Carlsbad, CA, USA) was added to PK-15 cells. At 72 hpt, the transfected cells were subjected to three freeze–thaw cycles to obtain cell lysates.

### 4.4. Indirect Immunofluorescence Assay and Western Blotting

For IFA, infected or transfected PK-15 cells were fixed with a cold solution of methanol/acetone (1:1) at −20 °C. Subsequently, the cells were blocked with 3% BSA (Roche, Mannheim, Germany). Primary antibodies, including anti-NS3, anti-PEDV S1 antibody (prepared in our laboratory), or anti-Flag antibody (Thermo Fisher Scientific, Waltham, MA, USA), were incubated with the cells for 90 min. Then, the cells were incubated with Alexa Fluor 488 goat anti-rabbit IgG or Alexa Fluor 594 goat anti-mouse IgG (Thermo Fisher Scientific, Waltham, MA, USA) for 60 min.

For Western blotting, PK-15 cells were infected with recombinant CSFV at an MOI of 0.1. At 48 hpi, supernatants and cell lysates were collected. The supernatants were concentrated 10-fold using an Amicon Ultra centrifugal filtration kit (Merck, Darmstad, Germany). The cell lysates or concentrated supernatants were subjected to analysis using specific antibodies, such as anti-VISA (prepared in our laboratory) or anti-PEDV S1 antibodies. For multiple probing, blots were stripped using the restore Western blot stripping buffer (Thermo Fisher Scientific, Waltham, MA, USA). A sensitive chemiluminescent horseradish peroxidase substrate (BioRad, München, Germany) was used to detect protein signals. Autoradiographic signals were detected by X-ray film (Fuji Medical X-ray Film, Tokyo, Japan).

### 4.5. Virus Titration and Plaque Assay

To determine the virus titer, a previously described method was followed [37]. Briefly, PK-15 cells were infected with the indicated viruses. At 72 hpi, the cells were harvested, and after three freeze–thaw cycles, the viral stocks were obtained by centrifugation. We added 100 µL of t10-fold serial dilutions of virus stock to PK-15 cells in a 96-well plate. At 72 hpi, the cells were fixed with a solution of methanol/acetone at −20 °C and an anti-NS3 antibody was used for an indirect immunofluorescence assay. The virus titers were determined using the TCID_50_ assay [60].

For plaque assay, SK-6 cells were infected with the indicated virus at an MOI of 0.001 for 1 h and then washed and covered with maintenance medium supplemented with 2% FBS and 1.5% carboxymethylcellulose. At 96 hpi, the cells were fixed with a solution of methanol/acetone using a Two-Step IHC Detection Reagent (ZSGB-BIO, Beijing, China) with an anti-NS3 antibody.

### 4.6. Luciferase Reporter Assay

To measure the luciferase activity for virus-mediated induction of the IFN-β promoter, SK-6 cells were infected with the indicated CSFV. After a 12 h incubation, the cells were co-transfected with 250 ng of pIFN-β-luc and 25 ng of pRL-TK. At 24 hpt, the cells were harvested and lysed for luciferase detection.

For plasmid-mediated activation of the IFN-β promoter, SK-6 cell monolayers were co-transfected with 500 ng of the indicated plasmid constructs, 250 ng of pIFN-β-luc, and 25 ng of pRL-TK using Lipofectamine^®^ 3000 reagent (Invitrogen). After 24 h of transfection, a dual-luciferase^®^ reporter assay (Promega, Madison, WI, USA) was used. The luciferase activities were reported as the relative expression levels of firefly luciferase (Fluc) to Renilla luciferase (Rluc). Each reporter gene assay was independently repeated at least three times.

### 4.7. Animal Experiments

For the immunization of rabbits, 15 New Zealand white rabbits (age 4 weeks) were randomized into five groups (n = 3). Rabbits in each group were inoculated intravenously with 10^4^ TCID_50_ recombinant chimeric CSFVs or DMEM [61]. All rabbits received boosters at the same dose at a 2-week interval. Bloods samples were collected from each rabbit 14 days after the booster immunization for the detection of antibody and cytokine concentrations.

For immunizing and challenging pigs, 22 specific CSFV- and PEDV-free piglets (age 5 weeks) were labeled by ear tags and randomly divided into 6 groups, including 2 groups for the recombinant CSFV immunization experiment (n = 5), 2 groups for the positive control (n = 3), and 2 groups for the negative control (n = 3). The piglets for recombinant CSFV experiments (2 groups) were intramuscularly vaccinated with 1 mL of 10^4.5^ TCID_50_/mL vC/SM3′UTR_N_-CARD/tPAsS1NCOE. Positive control group 1 was intramuscularly (i.m.) injected with 1 mL of 10^6^ TCID_50_/mL inactivated PEDV and group 2 with 1 mL of commercial C-strain vaccine (i.m.). The negative control group received 1 mL of DMEM (i.m.). After a 2-week interval, all groups received booster inoculation of the corresponding vaccines or DMEM. Fourteen days after the booster immunization, the experimental group received 1 mL (10^5.5^ TCID_50_/mL) of the intranasal virulent CSFV Shimen strain or 5 mL (10^5^ TCID_50_/mL) of the oral highly virulent PEDV GX750A strain for evaluation of anti-CSFV or anti-PEDV efficacy.

The experimental piglets’ rectal temperature and clinical signs were monitored daily. Oral and rectal swabs, as well as fecal samples, were collected for viral detection. The CSF and PED clinical scores were determined according to previously described methods (42, 43, 44). EDTA-treated blood and serum were sampled from the experimental pigs at 0, 7, 14, 21, and 28 dpv and 0, 3, 7, 14, and 21 dpc. White blood cells (WBCs) and platelets (PLTs) were counted using a hematological analyzer (SYSMEX, Kobe, Japan). The PED fecal clinical scores were graded as follows: 0 for normal stool, 1 for loose-consistency stool, 2 for semifluid-consistency stool, and 3 for watery diarrhea. Fecal samples from piglets challenged with virulent PEDV were collected on the indicated days for viral detection. The CSF clinical scores were graded as 0 for normal, 1 for slight alterations, 2 for distinct clinical signs, and 3 for severe CSF signs. Whole blood, oral swabs, and fecal swabs were collected from the experimental piglets on the indicated days (Table 1). The piglets that survived were euthanized at 21 dpc, and tissues from the tonsil, submandibular lymph node, spleen, and kidney were sampled for quantification of viral RNA copy numbers using RT-qPCR.

Animal experiments were approved by the Institutional Animal Care and Use Committee of Wuhan University (WDSKY0202107).

### 4.8. ELISA

To quantify CSFV-specific antibodies, a commercial ELISA kit from IDEXX (USA) was employed, as mentioned previously [62]. A positive result was defined as a ≥40% blocking value.

For the detection of PEDV-specific antibodies, 96-well plates were coated with purified inactivated PEDV (1.5 × 10^6^ TCID_50_/mL, 100 μL/well) and incubated overnight. After washing with PBST and blocking with 3% BSA (Roche, Mannheim, Germany), the wells were incubated with diluted rabbit serum (2000-fold) or diluted pig serum (400-fold), in duplicate, for 1 h at 37 °C. Following another round of washing, secondary antibodies (HRP-labeled goat anti-rabbit IgG or HRP-labeled goat anti-pig IgG or HRP-labeled goat anti-pig IgA, Thermo Fisher Scientific, Waltham, MA, USA) were added to the wells (1:10,000 dilution, 100 μL/well) and incubated for 1 h. After additional washes, TMB substrate (100 μL/well) was added and incubated for 20 min. After adding 2 M H_2_SO_4_, the reaction was stopped, and the levels of specific antibodies of experimental animals were expressed as OD_450_ values.

To detect IFN-γ in PBMCs, blood was sampled from vaccinated and control animals into heparin tubes by jugular venipuncture at 28 dpv. PBMCs were isolated within 4 h after collection using density-gradient centrifugation. The cell suspension was diluted (to 2 × 10^6^ cells/mL) and cultured in a 24-well culture plate (100 μL/well) with purified inactivated PEDV (1.5 × 10^5^ TCID_50_). After a 72 h culture, the culture supernatants were harvested for IFN-γ detection using a commercially available rabbit cytokine ELISA kit (Mlbio, Shanghai, China) or porcine cytokine ELISA kit (DuoSet ELISA, R&D Systems, Minneapolis, MN, USA).

### 4.9. Serum Neutralization Test

CSFV- and PEDV-specific virus-neutralizing (VN) titers were determined [63,64]. In summary, the serum samples were heat-inactivated. Serial dilutions of the serum samples (100 μL) were prepared and incubated with equal volumes of 100 TCID_50_ of either PEDV (GX750A) or CSFV (Shimen) at 37 °C for 1 h. For the determination of PEDV VN titers, Vero cells were infected with the serum–virus mixture in 96-well plates. After the 1 h incubation, the cells were replenished with fresh culture medium containing 5 μg/mL trypsin. Following a 7-day incubation period, the cells were examined for the presence of cytopathic effects (CPEs). The serum neutralization titer was expressed as the reciprocal of the highest serum dilution necessary to inhibit CPE. For the assessment of CSFV VN titers, the virus–serum mixture was transferred to PK-15 cells and incubated for 1 h. The cells were then provided with fresh culture medium containing 2% FBS. After a 72 h incubation, the cells were fixed, and immunofluorescence staining was performed. The virus-neutralizing antibody titer was reported as the reciprocal of the highest dilution necessary to prevent viral growth in 50% of the wells.

### 4.10. RT–qPCR

Viral RNA copy number was counted by RT-qPCR, as shown previously [62]. To extract total RNA, the RNApure Kit (Aidlab, Beijing, China) was used. Subsequently, the extracted RNA was used as a template for cDNA synthesis, which was performed using the ReverTra Ace qPCR RT Kit (Toyobo, Osaka, Japan). The resulting cDNA was then quantified using the SYBR^®^ Green Premix Pro Taq HS qPCR Kit (Accurate Biotech, Changsha, China) with specific primers (Appendix A). We followed the qPCR Premix composition guidelines and the amplification protocol provided by Accurate Biotech. Fluorescence signals were measured using a Bio-Rad CFX96. RNA copy numbers were determined based on duplicates of three independent experiments.

### 4.11. Statistical Analysis

The data were statistically analyzed by SPSS. Student’s *t* test was performed between two groups, and the nonparametric Kruskal–Wallis test or one-way ANOVA was performed for comparisons between multiple groups. *p* < 0.05 indicated statistical significance. Except where otherwise noted, data are shown as the mean ± SD. 

## Figures and Tables

**Figure 1 ijms-24-11954-f001:**
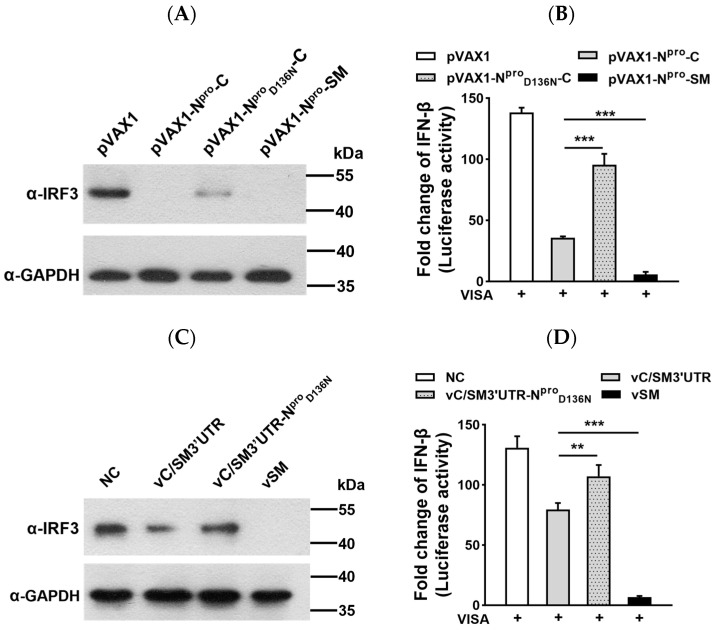
Regulation of IRF3 degradation and VISA-mediated IFN-β induction by the CSFV C-strain N^pro^_D136N_ variant. Plasmids or recombinant CSFVs harboring the N^pro^_D136N_ mutant (**A**,**C**) decreased IRF3 degradation and (**B**,**D**) reduced its inhibition of VISA-mediated IFN-β induction. SK-6 cells in 24-well plates were transfected with 500 ng of the indicated plasmid or infected with the indicated 0.1 multiplicity of infection (MOI) CSFV. For IRF3 analysis, the cells were harvested at 24 h post-transfection (hpt) or 48 h postinfection (hpi), and the samples were prepared for Western blotting using the anti-IRF3 mAb (Abcam). For IFN-β activation, at 12 h post-transfection with the indicated plasmid, or postinfection with the indicated CSFV, SK-6 cells were co-transfected with pIFN-β-luc, pRL-TK, and pVAX1-VISA. The luciferase activity was expressed as the relative expression levels of the Fluc/Rluc ratio. The comparisons were performed using Student’s *t* test or one-way ANOVA. ns, not significant; ** *p* < 0.01. *** *p* < 0.001.

**Figure 2 ijms-24-11954-f002:**
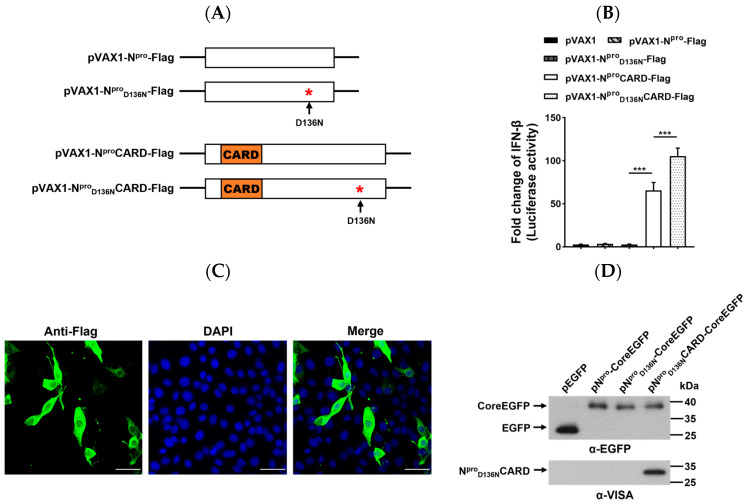
Construction and characteristics of CARD fused with wild-type and mutated N^pro^ proteins. (**A**) Schematic diagram of the eukaryotic plasmids expressing the N^pro^ protein and the chimera fused with the CARD of VISA. (**B**) Induction of IFN-β promoter activity. SK-6 cells were transfected with the indicated plasmids in the presence of pRL-TK plus pIFN-β-luc. At 24 hpt, Fluc and Rluc activities were assayed. (**C**) Expression of the fusion protein N^pro^_D136N_CARD. PK-15 cells were transfected with pVAX1-N^pro^_D136N_CARD. At 24 hpt, cells were subjected to indirect immunofluorescence assay (IFA) using anti-Flag mAb (Thermo Fisher Scientific, Waltham, MA, USA) and nuclear staining with DAPI. Stained cells were observed under a confocal microscope (Leica, Wetzlar, Germany). Scale bar represents 20 μm. (**D**) Autoprotease activity of the N^pro^ variant and the fusion protein. SK-6 cells were transfected with the indicated plasmids. At 48 hpt, the cells were harvested and lysed. The fusion proteins were analyzed by Western blotting using anti-EGFP or anti-VISA antibodies. The positions of the cleaved products are marked with arrows. The comparisons were performed using Student’s *t* test or one-way ANOVA. ns, not significant; *** *p* < 0.001.

**Figure 3 ijms-24-11954-f003:**
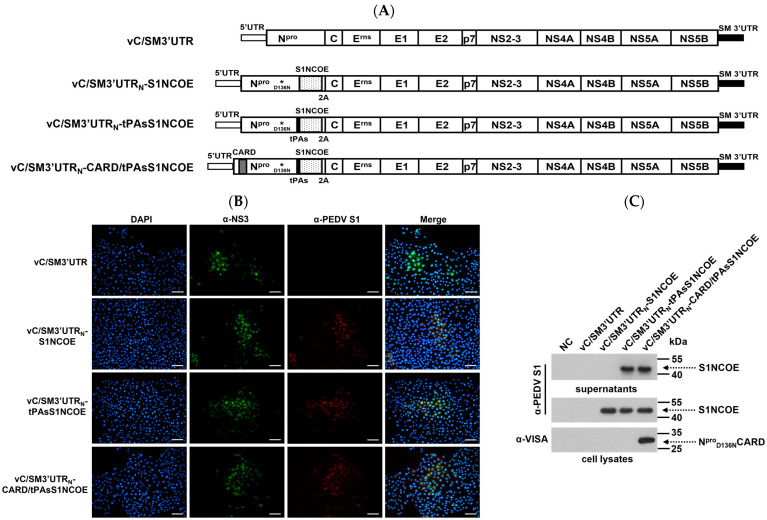
Rescue of recombinant CSFVs and heterogeneous protein expression. (**A**) Schematic diagrams of four recombinant CSFV constructs. (**B**) Identification of rescued viruses. PK-15 cells were infected with 0.01 MOI of the indicated virus. IFA was performed using anti-NS3 or anti-PEDV S1 antibodies and nuclear staining with DAPI. Scale bar represents 30 μm (**C**) Analyses of the heterogeneous proteins from four recombinant viruses. The culture supernatants and cells were collected from PK-15 cells infected with the indicated virus. Protein expression was probed by Western blotting.

**Figure 4 ijms-24-11954-f004:**
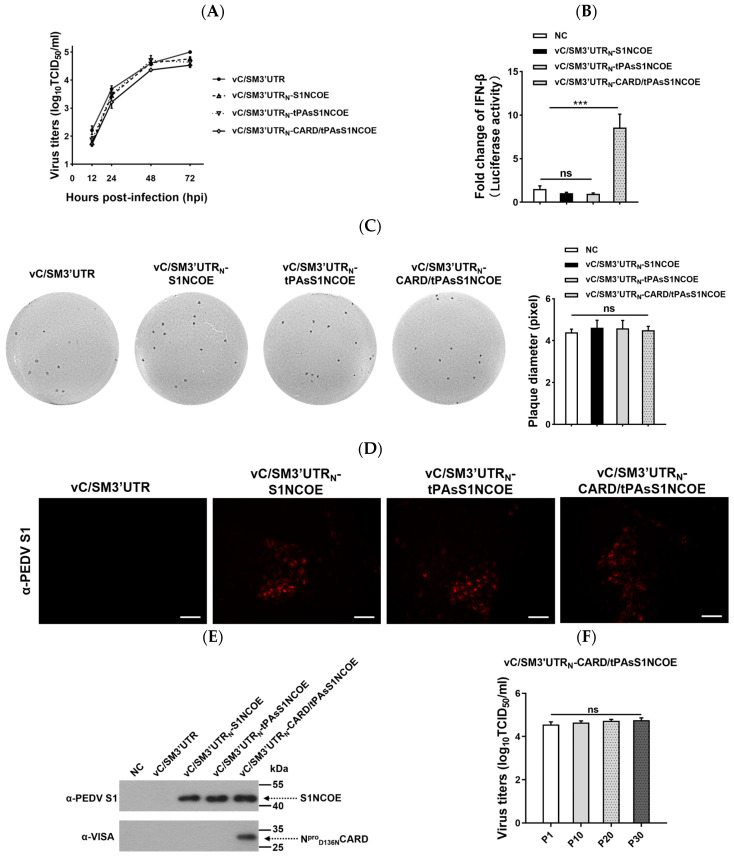
Biological characteristics of recombinant CSFVs. (**A**) Growth kinetics of the recombinant CSFVs. PK-15 cells were infected with 0.01 MOI of the indicated virus, respectively. The cells were collected and lysed at 12, 24, 48, and 72 hpi. Infectious viruses from the lysed supernatants were titrated on PK-15 cells. (**B**) IFN-β promoter activation induced by the recombinant CSFVs. SK-6 cells were infected with the indicated virus, and at 12 hpi the infected cells were co-transfected with pIFN-β-luc and pRL-TK. At 24 hpt, the cells were harvested and lysed for luciferase detection. (**C**) Plaque morphology of four recombinant CSFVs. SK-6 cells were infected with the indicated virus. At 96 hpi, the infected cells were immunohistochemically stained using anti-NS3 antibody. (**D**,**E**) Heterogeneous protein expression of the passaged viruses. PK-15 cells infected with the indicated virus were serially passaged separately. The 30th infected PK-15 cells were analyzed by IFA using an anti-PEDV S1 antibody. (**D**) Scale bar represents 30 μm; (**E**) the 30th infected cells were harvested and lysed for Western blotting analysis using anti-PEDV S1 or anti-VISA antibodies. (**F**) Virus titers of the passaged chimeric CSFV. The 1st (P1), 10th (P10), 20th (P20), and 30th (P30) passaged cells infected with vC/SM3′UTR_N_-CARD/tPAsS1NCOE were harvested for virus titration. The comparisons were performed using Student’s *t* test or one-way ANOVA. ns, not significant; *** *p* < 0.001.

**Figure 5 ijms-24-11954-f005:**
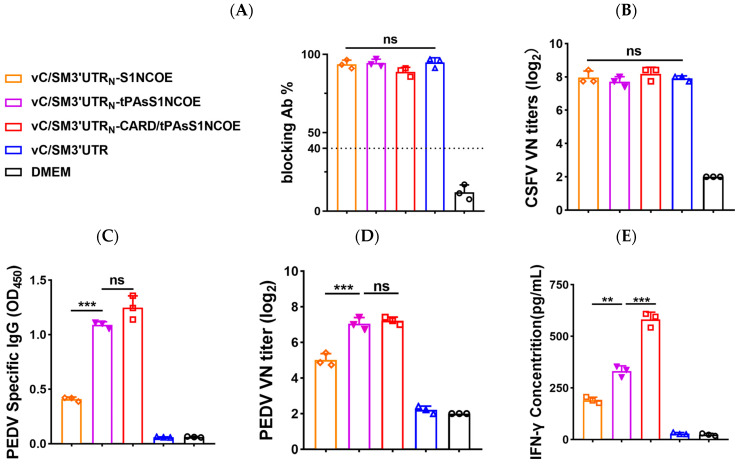
Humoral immune response and PEDV-specific IFN-γ production induced by recombinant CSFVs in rabbits. Rabbits were inoculated twice intravenously with the indicated virus at a 2-week interval (n = 3). DMEM was used as a negative control (n = 3). The serum of the rabbits was collected at 14 days postbooster, and virus-specific antibodies and neutralizing antibodies were determined by ELISA and neutralizing assay, respectively. PEDV-specific IFN-γ in peripheral blood lymphocytes collected 14 days postbooster was measured using a commercially available cytokine ELISA kit (Mlbio, Shanghai, China). (**A**) CSFV-specific antibody titers. (**B**) CSFV-neutralizing antibody titers. (**C**) PEDV-specific IgG concentration. (**D**) PEDV-neutralizing antibody titers. (**E**) PEDV-specific IFN-γ concentration. The comparisons were performed using Student’s *t* test, one-way ANOVA or the nonparametric Kruskal–Wallis test. ns, not significant; ** *p* < 0.01. *** *p* < 0.001.

**Figure 6 ijms-24-11954-f006:**
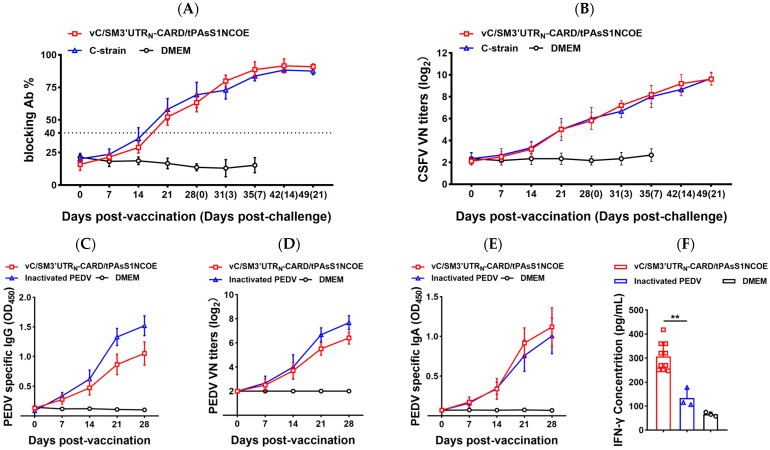
Humoral immune response and PEDV-specific IFN-γ production induced by recombinant CSFVs in pigs. Specific CSFV- and PEDV-free piglets aged 5 weeks were intramuscularly vaccinated twice with the indicated inoculum at a 2-week interval. Sera of the piglets were collected at the indicated timepoints, and virus-specific antibodies and neutralizing antibodies were determined by ELISA and the neutralizing assay. PEDV-specific IFN-γ in PBMCs collected 14 days postbooster was measured. (**A**) CSFV-specific antibody titers. (**B**) CSFV neutralizing antibody titers. (**C**) PEDV-specific IgG concentration. (**D**) PEDV neutralizing antibody titers. (**E**) PEDV-specific IgA concentration. (**F**) PEDV-specific IFN-γ concentration. The comparisons of the values were performed using Student’s *t* test. ** *p* < 0.01.

**Figure 7 ijms-24-11954-f007:**
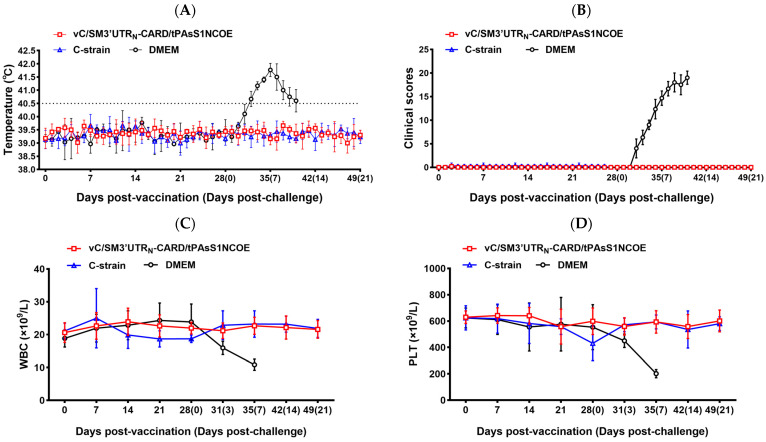
Immune protection of vaccinated pigs against virulent CSFV challenge. (**A**) Daily rectal temperature of vaccinated piglets and the following virulent CSFV challenge. (**B**) Clinical scores of vaccinated pigs following virulent CSFV challenge as previously described by Mittelholzer. (**C**) WBC counts of vaccinated piglets following virulent CSFV challenge. (**D**) PLT counts of vaccinated piglets following virulent CSFV challenge.

**Table 1 ijms-24-11954-t001:** CSFV RNA copies in the whole blood, oral, and fecal swabs detected by RT–qPCR.

Inocula	Sample	N.E.P *^a^*	Days Post-Vaccination (Days Post-Challenge)								
			0	7	14	21	28(0)	31(3)	35(7)	42(14)	49(21)
vC/SM3′UTR_N_-CARD/tPAsS1NCOE	Blood	158	– *^b^*	–	–	–	–	–	–	–	–
		169	–	–	–	–	–	–	–	–	–
		181	–	–	–	–	–	–	–	–	–
		182	–	–	–	–	–	–	–	–	–
		192	–	–	–	–	–	–	–	–	–
	Oral swab	158	–	–	–	–	–	–	–	–	–
		169	–	–	–	–	–	–	–	–	–
		181	–	–	–	–	–	–	–	–	–
		182	–	–	–	–	–	–	–	–	–
		192	–	–	–	–	–	–	–	–	–
	Fecal swab	158	–	–	–	–	–	–	–	–	–
		169	–	–	–	–	–	–	–	–	–
		181	–	–	–	–	–	–	–	–	–
		182	–	–	–	–	–	–	–	–	–
		192	–	–	–	–	–	–	–	–	–
C-strain	Blood	174	–	–	–	–	–	–	–	–	–
		186	–	–	–	–	–	–	–	–	–
		209	–	–	–	–	–	–	–	–	–
	Oral swab	174	–	–	–	–	–	–	–	–	–
		186	–	–	–	–	–	–	–	–	–
		209	–	–	–	–	–	–	–	–	–
	Fecal swab	174	–	–	–	–	–	–	–	–	–
		186	–	–	–	–	–	–	–	–	–
		209	–	–	–	–	–	–	–	–	–
DMEM	Blood	157	–	–	–	–	–	1.41 × 10^3^	1.66 × 10^7^	/ *^c^*	/
		204	–	–	–	–	–	1.83 × 10^3^	1.72 × 10^6^	/	/
		208	–	–	–	–	–	3.24 × 10^2^	1.49 × 10^5^	/	/
	Oral swab	157	–	–	–	–	–	1.26 × 10^3^	1.27 × 10^7^	/	/
		204	–	–	–	–	–	2.12 × 10^3^	4.56 × 10^6^	/	/
		208	–	–	–	–	–	3.95 × 10^3^	1.45 × 10^6^	/	/
	Fecal swab	157	–	–	–	–	–	1.70 × 10^3^	1.24 × 10^5^	/	/
		204	–	–	–	–	–	1.26 × 10^3^	2.01 × 10^5^	/	/
		208	–	–	–	–	–	5.41 × 10^2^	9.37 × 10^4^	/	/

*^a^* number of experimental piglets; *^b^* not detected. *^c^* piglet died.

**Table 2 ijms-24-11954-t002:** CSFV RNA copies in selected tissues detected by RT–qPCR.

Inocula	vC/SM3′UTR_N_-CARD/tPAsS1NCOE (n = 5)					C-Strain (n = 3)			DMEM (n = 3)		
N.E.P *^a^*	158	169	181	182	192	174	186	209	157	204	208
Tissues											
Tonsil	– *^b^*	–	–	–	–	–	–	–	3.56 × 10^7^	4.66 × 10^6^	3.77 × 10^6^
Submandibular lymph node	–	–	–	–	–	–	–	–	6.26 × 10^7^	6.02 × 10^6^	4.47 × 10^6^
Spleen	–	–	–	–	–	–	–	–	4.73 × 10^6^	7.19 × 10^5^	6.01 × 10^5^
Kidney	–	–	–	–	–	–	–	–	4.94 × 10^5^	5.95 × 10^4^	9.87 × 10^4^

*^a^* number of experimental piglets; *^b^* not detected.

**Table 3 ijms-24-11954-t003:** PEDV and clinical scores in feces of vaccinated piglets following virulent PEDV challenge.

Inocula		vC/SM3′UTR_N_-CARD/tPAsS1NCOE (n = 5)					Inactivated PEDV (n = 3)				DMEM (n = 3)			
N.E.P *^a^* 156		160	205	210	217		161	202	203		152	155	214	
dpc *^b^*	Viral RNA Copies					CS *^c^*	Viral RNA Copies			CS	Viral RNA Copies			CS
0	– *^d^*	–	–	–	–	0/0/0/0/0	–	–	–	0/0/0	–	–	–	0/0/0
3	–	–	–	9.81 × 10^2^	2.94 × 10^3^	0/0/0/0/1	1.01 × 10^3^	–	1.94 × 10^3^	0/0/1	9.59 × 10^5^	6.94 × 10^6^	6.79 × 10^5^	2/3/2
5	–	1.87 × 10^3^	2.98 × 10^3^	3.56 × 10^3^	4.91 × 10^3^	0/0/1/1/1	1.51 × 10^4^	2.75 × 10^4^	3.63 × 10^4^	1/1/1	7.17 × 10^7^	1.01 × 10^8^	9.51 × 10^6^	3/3/3
7	–	–	–	1.61 × 10^3^	1.61 × 10^3^	0/0/0/1/1	–	9.08 × 10^3^	1.10 × 10^3^	0/1/0	1.06 × 10^6^	7.49 × 10^6^	5.87 × 10^5^	2/3/2
9	–	–	–	–	–	0/0/0/0/0	–	–	–	0/0/0	1.09 × 10^6^	4.31 × 10^6^	6.95 × 10^3^	2/2/1
11	–	–	–	–	–	0/0/0/0/0	–	–	–	0/0/0	3.64 × 10^5^	6.69 × 10^5^	8.10 × 10^3^	1/1/0
14	–	–	–	–	–	0/0/0/0/0	–	–	–	0/0/0	7.88 × 10^3^	2.97 × 10^4^	1.98 × 10^3^	0/0/0

*^a^* number of experimental piglets; *^b^* days post-challenge; *^c^* clinical score for fecal consistency; *^d^* not detected.

## Data Availability

The data presented in this study are available on request from the corresponding author.

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
