# Peer review of "A Genetically Engineered Bivalent Vaccine Coexpressing a Molecular Adjuvant against Classical Swine Fever and Porcine Epidemic Diarrhea"

_ijms, 2023, doi:10.3390/ijms241511954_

Round 1
Reviewer 1 Report
To this reviewer, the manuscript is well-written, and the research is original and clearly presented. The topic deserves attention, and the developed vaccine approach may be useful.
This reviewer identified no major issues or missing controls.
This reviewer identified no significant issue with the language.
Reviewer 2 Report
The manuscript by Wang et al describes the use of a bivalent vaccine construct against CSF and PDE. This study offers a novel platform to target two viruses while inducing humoral immunity. The manuscript is well written and organized. The authors highlight specific features of the vaccine's efficacy. Please see my comments below:
1. When describing the nature of the constructs, the authors should provide more information to make it clear to the reader. For instance, the construct VC/SM3’UTR-N is not clearly indicated that its 3'UTR was replaced with the one from another viral strain. This should be clearly stated very early in the manuscript (Figure1). We (reader) don't get to know the nature of this construct until we reach Figure 3.
2. In Figure 3, the authors did not indicate the MOI used for IFA.
3. In Figure 4A, titers are very similar for all viral constructs. However, when evaluating IFN-B expression, virus C/SM3’UTRN-CARD/tPAsS1NCOE showed the highest induction of IFN-B. Is this due to the difference in cells used? Aren't PK-15 cells sensitivity to IFN or able to produce IFN? The authors should also explain why they decide to use PK-15 cells for passing the virus, shouldn't they use other cell lines, perhaps primary cells to determine if the virus is genetically stable under more physiologically relevant conditions?
4. Only RNAemia was shown after challenge. Why did the authors not show virus isolation for at least the control groups? If the animals were protected and inducting humoral responses, this means that the virus should have been replicating. Why then, there was no indication of viremia or local detection of infectious virus in the tonsils, LN, etc. The authors should expand on that.
Reviewer 3 Report
I reviewed the manuscript entitled: “A genetically engineered bivalent vaccine co-expressing a molecular adjuvant against classical swine fever and porcine epidemic diarrhea” In this manuscript authors present the successful development of a vaccine candidate to protect pigs against both classical swine fever and porcine epidemic diarrhea.
Overall, I think it is a valuable work. It is a well detailed paper and the methodology used by the authors supports the results of this study.
My suggestion to improve the quality of this study, is to improve the discussion section. I suggest adding more information about the genetic variability of this viruses and how this vaccine may perform using other strains from different genetic origin. Discuss more about future validations. Include more information comparing the performance of this vaccine with previous ones published for these diseases.
A question from the results section is about the lack of detection of infectious virus, after the challenge. I suggest including information regarding the equivalence between copy numbers or CT values and infectious titers.
Round 2
Reviewer 3 Report
I like to thank the authors for their responses. At this point, I don't have more concerns about this study.